# COMPREHEND AND TALK: TEXT TO SPEECH SYNTHESIS VIA DUAL LANGUAGE MODELING

## ABSTRACT

Existing Large Language Model (LLM) based autoregressive (AR) text-to-speech (TTS) systems, while achieving state-of-the-art quality, still face critical challenges. The foundation of this LLM-based paradigm is the discretization of the continuous speech waveform into a sequence of discrete tokens by neural audio codec. However, single codebook modeling is well suited to text LLMs, but suffers from significant information loss; hierarchical acoustic tokens, typically generated via Residual Vector Quantization (RVQ), often lack explicit semantic structure, placing a heavy learning burden on the model. Furthermore, the autoregressive process is inherently susceptible to error accumulation, which can degrade generation stability. To address these limitations, we propose CaT-TTS, a novel framework for robust and semantically-grounded zero-shot synthesis. First, we introduce S3Codec, a split RVQ codec that injects explicit linguistic features into its primary codebook via semantic distillation from a state-of-the-art ASR model, providing a structured representation that simplifies the learning task. Second, we propose an "Understand-then-Generate" dual-Transformer architecture that decouples comprehension from rendering. An initial "Understanding" Transformer models the cross-modal relationship between text and the audio's semantic tokens to form a high-level utterance plan. A subsequent "Generation" Transformer then executes this plan, autoregressively synthesizing hierarchical acoustic tokens. Finally, to enhance generation stability, we introduce Masked Audio Parallel Inference (MAPI), a nearly parameter-free inference strategy that dynamically guides the decoding process to mitigate local errors. Extensive experiments demonstrate that the synergy of our principled architecture and semantically-aware codec allows CaT-TTS to achieve new state-of-the-art performance in zero-shot voice cloning, with MAPI providing a measurable boost in generation robustness on benchmark datasets. Project page: https://anonymous.4open.science/r/CaT-TTS-66A1.

## 1 INTRODUCTION

Large Language Model (LLM) based autoregressive (AR) models have achieved state-of-the-art quality in zero-shot Text-to-Speech (TTS) with discrete audio representations (Wang et al., 2023; Du et al., 2024a;b; Anastassiou et al., 2024). With a few seconds of audio prompt, current TTS models are able to synthesize speech for any given text and mimic the speaker of the audio prompt. Contrary to NAR models (Chen et al., 2024b; Le et al., 2023), the sequential nature of AR models, where each acoustic token is conditioned on all its predecessors, naturally captures the long-range temporal dependencies essential for rendering intricate intonation, rhythm, and emotional nuance. This sequential process synergizes perfectly with the in-context learning (ICL) capabilities of LLMs (Ye et al., 2025b; Wang et al., 2025b), providing a powerful mechanism for propagating the fine-grained acoustic characteristics of a voice prompt throughout a newly synthesized utterance.

Despite the remarkable progress in LLM-based zero-shot TTS, several fundamental challenges persist. The foundation of this LLM-based paradigm is the discretization of the continuous speech waveform into a sequence of discrete tokens, a task handled by a neural audio codec (Kreuk et al., 2022; Copet et al., 2023). Semantic tokens, typically derived from discretized self-supervised learning (SSL) models, are considered to exhibit high alignment with text while leading to poor reconstruction (Du et al., 2024a; Ye et al., 2025a; Gong et al., 2025). In contrast, acoustic tokens often

derived from speech codecs trained through residual vector quantization GAN (RVQ-GAN), are recognized for capturing the details of the audio waveform, enabling high-quality synthesis, but lack explicit semantic grounding, forcing the LLM to learn the complex mapping from text to raw acoustic properties from scratch (Défossez et al., 2024; Kumar et al., 2023; Han et al., 2025). We assume that a better audio tokenizer should contain rich semantic information to facilitate an easy understanding of audio content, thus reducing the language model's burden in interpreting tokens, and contains acoustic information for speech reconstruction. For better linguistic understanding and acoustic reconstruction, inspired by Mimi codec and SpeechTokenizer (Défossez et al., 2024; Zhang et al., 2023a), we propose S3Codec, a split residual vector quantization speech codec with semantic distillation. However, rather than using SSL models, we adopt a pretrained state-of-the-art ASR model for semantic distillation, which we assume brings more explicit linguistic features.

We argue that speech synthesis is fundamentally an information-increasing process, where a thorough understanding of the source conditions is a prerequisite for accurate and effective generation (Chu et al., 2023; Xu et al., 2025; Xie & Wu, 2024). To embody this principle, we propose CaT-TTS, a novel "Comprehend-and-Talk" text-to-speech framework, realized through a dual-transformer architecture that explicitly decouples contextual comprehension from acoustic rendering. Our first module, the Semantic Transformer, operates autoregressively on the semantic level. Its sole purpose is to model the rich interplay between the input text and the core semantic content of the voice prompt, building a holistic high-level representation, a latent "plan" for the entire utterance. Following this, our second module, the Acoustic Transformer, takes this contextual plan as its foundation and executes the synthesis. It generates the detailed acoustic tokens autoregressively. This design allows the model first to understand "what" and "how", and then generates the "sound", which dramatically reduces the modeling burden at each step, leading to more coherent and expressive output.

While our architectural design provides a more stable foundation, the challenge of long sequence lengths in speech remains (Zhang et al., 2023b; Le et al., 2023). Even with our proposed high compression ratio codec, which significantly shortens the acoustic token sequences, the risk of error accumulation persists in any AR system. To overcome this challenge, inspired by Classifier-Free Guidance (CFG) in diffusion models (Ho & Salimans, 2022) and Parallel Scaling Laws (Chen et al., 2025), we introduce Masked Audio Parallel Inference (MAPI). It constructs parallel computing streams with different masked audio tokens and aggregates these streams adaptively with learnable weights. This technique acts as a corrective mechanism, steering the model back on track when it begins to "hallucinate" and ensuring robust output.

In summary, we propose a novel zero-shot TTS system CaT-TTS powered by S3Codec. S3Codec encompasses acoustic and semantic information with low bit rates. Based on S3Codec, Cat-TTS embodies an understand and then generate rules via a dual language modeling strategy. To mitigate the error accumulation problem in audio language models, we introduce Masked Audio Parallel Inference strategy, which is beneficial for more robust token generation. Extensive experiments have shown that CaT-TTS has achieved a comparable or superior quality to existing models in terms of speech quality, similarity, and intelligibility.

## 2 RELATED WORK

**Speech Tokenization.** The success of autoregressive language models has spurred progress in speech LLMs, where speech tokenizers are essential for converting continuous signals into discrete tokens. Speech tokenizers are typically categorized as acoustic or semantic (Wang et al., 2025a; Yang et al., 2025). Acoustic tokens, optimized for signal reconstruction, capture detailed acoustic features beneficial for generation, but perform poorly on understanding tasks like ASR. Previous semantic tokenizers can be trained in two ways: (1) applying clustering or VQ to the representations of self-supervised learning models (Zhang et al., 2023a; Défossez et al., 2024). (2) applying a VQ layer to the intermediate layer of ASR models (Du et al., 2024a;b). These semantic tokenizers typically use a single codebook, have a simple architecture, are rich in linguistic information, and are well-suited for LLMs. However, finer-grained acoustic details such as pitch and prosody, are lost, resulting in poor performance on generation tasks (Łajszczak et al., 2024; Betker, 2023). An alternative for audio tokenization is to use multi-codebook residual vector quantization (RVQ). In RVQ, an audio frame is represented by a sum of vectors from several quantizers, allowing for

high-fidelity reconstruction over a range of bitrates by capturing details that single-codebook models often miss (Kumar et al., 2023; Défossez et al., 2022; Zeghidour et al., 2021). To align residual speech codec tokens with large text models, recent efforts have explored modeling both semantic and acoustic features simultaneously. SpeechTokenizer (Zhang et al., 2023a) enhances the RVQ-GAN paradigm with semantic distillation to guide the first layer of RVQ to align with a teacher SSL model. X-codec (Ye et al., 2025a) proposes an X-shaped structure where each layer of RVQ contains semantic and acoustic information. Mimi (Défossez et al., 2024) argues that distilling semantic information into the first level of a single RVQ will trade the auido quality restoration performance of the residual codebooks. Similar to Mimi, we propose S3Codec: a Split RVQ Speech Tokenizer with Semantic Distillation. Unlike Mimi, we adopt DAC architecture with pretrained Whisper for semantic distillation. This approach allows S3Codec to have good acoustic restoration ability and stronger linguistic information.

**LLM-based Zero-Shot TTS.** Inspired by the success of LLM, several recent works adopt language models to model text-to-speech (TTS) tasks (Chen et al., 2024a; Kharitonov et al., 2023; Meng et al., 2024). The LLM-based TTS systems are typically trained on tens of thousands of hours of speech data and have hundreds of millions of parameters, hence can leverage the emergent abilities of LLMs like in-context learning to enable zero-shot TTS. VALL-E pioneered treating TTS as a conditional language modeling problem by converting waveforms into neural codec tokens. Spear-TTS (Kharitonov et al., 2023) integrates multiple AR models to support multispeaker TTS with minimal supervision. Many systems use a single discrete codebook to quantize semantic features (Wang et al., 2025b; Ye et al., 2025b). Although simple, this bottleneck loses fine acoustic detail (Han et al., 2025). Recent TTS systems have often combined an AR language model with additional components (Du et al., 2024a), such as diffusion, to generate more natural, controllable speech when trained on large datasets. While these methods can produce high-quality results, most of them neglect the interactive understanding of speech and text modalities, instead requiring continuous and fine-grained acoustic features for supplementation. Storing and processing such large-scale features is prohibitive, hindering training on hundreds of billions of tokens. In contrast, our approach utilizes a dual-autoregressive structure powered by a split RVQ discretization technique, with the first semantic transformer for modality understanding and the second acoustic transformer for acoustic information generation based on the context guide produced by the semantic transformer. This understand-then-generate paradigm fits the natural flow of speech, takes advantage of the context learning of LLMs, and avoids the need for additional acoustic features for supplementary reconstruction.

## 3 METHOD

### 3.1 S3CODEC: SPLIT RVQ WITH SEMANTIC DISTILLATION FOR SPEECH TOKENIZER

To discretize waveforms into audio tokens, we introduce S3Codec, a neural audio codec that operates as an autoencoder with a discrete bottleneck. Figure 2 shows the architecture. Based on the DAC architecture (Kumar et al., 2023), the encoder projects a single-channel waveform $\mathbf{x} \in \mathbb{R}^T$ to a latent representation $\mathbf{A} = \text{enc}(\mathbf{x}) \in \mathbb{R}^{L \times D}$ by cascading residual convolutional blocks that interleave dilated and strided convolutions along with Snake nonlinearities and weight normalizaton, and Quantizer quantize the latent representation to discrete representations $\mathbf{C} \in \mathbb{R}^{K \times L \times D}$ where $L$ represents the length of encoded tokens, $K$ represents the number of codebooks and $D$ represents the dimension of codebook. Similarly to SpeechTokenizer and Mimi, we distill semantic information into the first level of RVQ. However, instead of using SSL models like HuBERT (Hsu et al., 2021) as a semantic teacher, we adopt Whisper (Radford et al., 2023), a state-of-the-art model for automatic speech recognition and speech translation whose hidden representation contains rich explicit linguistic features. Mimi (Défossez et al., 2024) found that, while distillation significantly improves the phonetic discriminability of the first quantizer, it also negatively affects the audio quality. To address the issue, we split the RVQ layers in a way similar to Mimi. Rather than a single RVQ with $K$ levels, we distill the semantic information into a plain VQ and apply an RVQ with $K-1$ levels in parallel; thus the constraint of acoustic information being conserved in the residual of the semantic quantizer is removed.

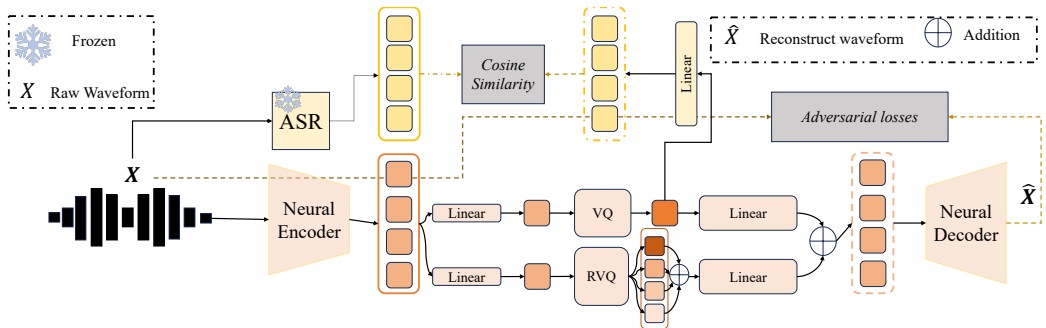

Figure 1: S3Codec architecture overview.

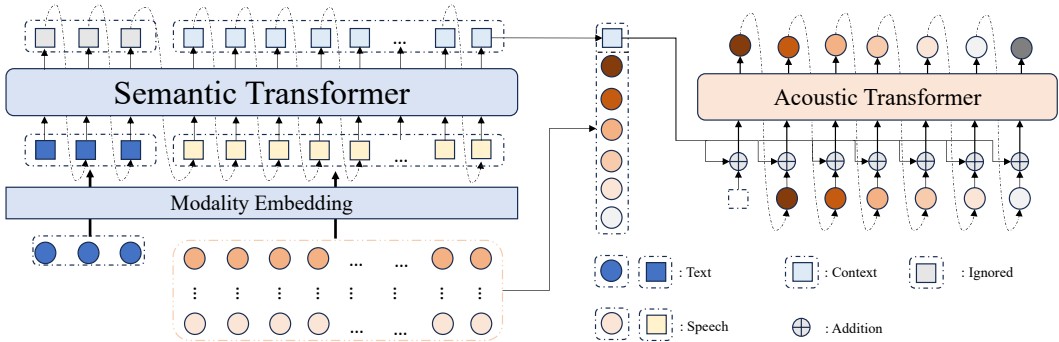

Figure 2: An overview of CaT-TTS architecture. Semantic Transformer models the temporal context information, while Acoustic Transformer models the acoustic information from coarse to fine.

### 3.1.1 TRAINING OBJECTIVE

S3Codec is trained with the combination of reconstruction, semantic distillation and adversarial losses. Reconstruction and adversarial losses can be found in Appendix E. For semantic distillation task, we calculated the cosine distance between the output of the first quantizer and the transformed Whisper embeddings, which is denoted as $\cos(\cdot)$, to perform distillation. Formally, the distillation loss is defined as follows:

$$\mathcal{L}_{distill} = 1 - \frac{1}{L}\sum_{t=1}^{L}\cos(\mathbf{C}_t^0, \text{Proj}(\mathbf{E}^{\mathcal{S}})_t), \tag{1}$$

where $\mathbf{C}_t^0 \in \mathbb{R}^D$ represents the first encoded embeddings for the frame $t$, $\mathbf{E}^{\mathcal{S}} \in \mathbb{R}^{L_{\mathcal{S}} \times D_{\mathcal{S}}}$ represents the semantic embeddings obtained from the Whisper Encoder, $\text{Proj}(\cdot) : \mathbb{R}^{L_{\mathcal{S}} \times D_{\mathcal{S}}} \to \mathbb{R}^{L \times D}$ represents the projection operation that maps whisper latent embedding to the space of audio embedding, $L_{\mathcal{S}}$ represents the length of semantic frames, and $\text{Proj}(\mathbf{E}^{\mathcal{S}})_t$ represents the projected whisper embedding for frame $t$. The details of the overall training objective are listed in the Appendix C.

### 3.2 DUAL LANGUAGE MODELING OF AUDIO TOKENS

#### 3.2.1 PROBLEM FORMULATION

Given a dataset $\mathcal{D} = \{\mathbf{x}, \mathbf{y}\}$, where $\mathbf{y}$ is an audio sample and $\mathbf{x}$ is the corresponding text transcription. We use a pre-trained neural codec model to encode each audio sample into discrete codes, denoted as $\text{S3Codec}(\mathbf{y}) = \mathbf{A} \in \mathbb{R}^{K \times L}$, where $K$ represents the number of codebooks, and $L$ is the downsampled utterance length. $\mathbf{A}_t \in \mathbb{R}^K$ represents the $K$ codes for frame $t$ and $\mathcal{A}_t^k$ represents the code for the $k$-th codebook of frame $t$. Mathematically, given the text prompt $\mathcal{T}$ and the speech prompt $\tilde{\mathbf{A}}$, our target is to train a neural language model to generate the discrete code matrix $\mathbf{A}$ with the optimization objective of maximizing the distribution:

$$\mathbb{P}(\mathbf{A}|\mathcal{T}, \tilde{\mathbf{A}}). \tag{2}$$

To build such a model, we propose a dual auto-regressive Transformer modeling framework. The dual auto-regressive (AR) Transformer models the residual vector quantization (RVQ) output as a two-level autoregressive process, operating first along the temporal axis and subsequently across codebooks. The core intuition behind this design is to preserve both the causal nature of speech generation and the hierarchical refinement characteristic of RVQ. We denote the first transformer as the semantic transformer, following the causal nature of speech generation and context learning, while the second transformer is the acoustic transformer, modeling the acoustic feature in a coarse-to-fine manner.

### 3.2.2 SEMANTIC TRANSFORMER

The semantic transformer functions as a thinker responsible for processing and understanding the text and the audio modality, and generating high-level representations. Mathematically, let $\mathcal{T} \in \mathbb{R}^M$ represent the tokenized textual prompt, $\mathbf{A} \in \mathbb{R}^{K \times L}$ represent the corresponding speech, and $\mathbf{A}^i \in \mathbb{R}^L, i = \{0, \cdots, K-1\}$ represent the speech codes in the i-th codebook, where $M$ represents the length of the encoded text token and $L$ represents the length of the encoded speech token. Given tokenized text prompt and encoded prompt audio codes, the semantic transformer learns the linguistic features of the text $\mathcal{T}$ and the discrete acoustic representation of the prompt audio $\tilde{\mathbf{A}}$ and outputs a latent feature $\mathbf{H}^{ctx}$ as a guide for the generation of subsequent speech tokens. The optimization objective of the semantic transformer is maximizing the distribution:

$$\mathbb{P}(\mathbf{H}^{cxt}|\mathcal{T}, \tilde{\mathbf{A}}; \theta_{\mathcal{S}}) = \prod_{t=1}^{L} \mathbb{P}(\mathbf{H}_t^{cxt}|\mathcal{T}, \mathbf{H}_{<t}^{ctx}, \tilde{\mathbf{A}}; \theta_{\mathcal{S}}). \tag{3}$$

**Speech Token Sequence Modeling.** To be able to inject discrete speech representations into LLM, some research proposes to use a single codebook codec to make the speech modality well adapted in the way of text tokens. CaT-TTS fits the RVQ paradigm and, specifically, the multiple codebook information at each time step will be aggregated as the speech representation of the current time step. Thus, at each time step $t$, the audio representation can be formulated as $\mathbf{S}_t = \sum_{i=0}^{K-1} \mathcal{A}_t^i$, where $\mathcal{A}_t^i$ represents the $i$-th encoded representation for frame t.

**Next Token Embedding Prediction.** In order to inject RVQ speech representation into LLM, we sum the codebook dimensions of the multi-codebook parallel sequence. Aggregation brings rich linguistic and acoustic content to the semantic transformer; however, the speech representation of each time step is no longer a quantitative representation. To solve this problem, we propose direct embedding prediction. Instead of predicting discrete token IDs and computing cross-entropy loss, we directly predict the next embedding vector in the continuous semantic space and optimize using Mean Squared Error Loss between predicted and target embeddings. Specifically, our model learns to predict the next semantic embedding as $\mathbf{H}_{t+1}^{ctx} = \theta_{\mathcal{S}}(\mathbf{H}_1^{ctx}, \mathbf{H}_2^{ctx}, ..., \mathbf{H}_t^{ctx})$, where $\mathbf{H}_t^{ctx}$ represents the continuous semantic embedding at position $t$. To be more task-specific, we denote $\mathbf{H}^{ctx} \doteq (\mathbf{T} \oplus \mathbf{S})$, where $\mathbf{T}$ represents the text modality, $\mathbf{S}$ represents the audio modality, and $\oplus$ represents the concatenate operation. We split the high-level representation and focus on speech modality; thus the optimization objective can be formulated as follows:

$$\mathbb{P}(\mathbf{H}^{cxt}|\mathcal{T}, \tilde{\mathbf{A}}; \theta_{\mathcal{S}}) = \mathbb{P}(\mathbf{S}|\mathcal{T}, \tilde{\mathbf{A}}; \theta_{\mathcal{S}}) = \prod_{t=1}^{L_{|S|}} \mathbb{P}(\mathbf{S}_t|\mathcal{T}, \mathbf{S}_{<t}, \tilde{\mathbf{A}}; \theta_{\mathcal{S}}), \tag{4}$$

where $L_{|S|}$ represents the length of speech frames, as the text tokens are ignored. To achieve this, we replace the standard cross-entropy loss with MSE loss to handle continuous targets:

$$\mathcal{L}_{ctx} = -\sum_{t=1}^{L_{|S|}} \log \mathbb{P}(\mathbf{S}_t|\mathcal{T}, \mathbf{S}_{<t}, \tilde{\mathbf{A}}; \theta_{\mathcal{S}}) \rightarrow \mathcal{L}_{ctx} = \sum_{t=1}^{L_{|S|}} ||\mathbf{S}_t - \theta_{\mathcal{S}}(\mathbf{S}_{<t}, \mathcal{T}, \tilde{\mathbf{A}})||_2. \tag{5}$$

### 3.2.3 ACOUSTIC TRANSFORMER

The purpose of the acoustic transformer is to reconstruct discrete speech representations from coarse-grained to fine-grained based on the learned preceding text and speech modal information. The optimization objective of the acoustic transformer is maximizing the following distribution:

$$\mathbb{P}(\mathbf{A}_t|\mathbf{S}_t; \theta_{\mathcal{A}}) = \prod_{k=0}^{K-1} \mathbb{P}(\mathcal{A}_t^k|\mathcal{A}_t^{<k}, \mathbf{S}_t; \theta_{\mathcal{A}}). \tag{6}$$

Table 1: Objective Evaluation Metrics for Comparison with Baseline Codecs. S-T represents SpeechTokenizer for simplicity.

| Tokenizer | CB | Nq | FR | BR (bps) | PESQ ↑ | STOI ↑ | STFT ↓ | Mel ↓ | SIM ↑ |
|-----------|-----|-----|--------|----------|--------|--------|--------|-------|-------|
| Encodec | 1024 | 8 | 75Hz | 6k | 2.76 | 0.94 | 0.11 | 2.13 | 0.89 |
| DAC-8 | 1024 | 8 | 75Hz | 6k | **3.46** | **0.95** | 0.06 | 2.02 | **0.96** |
| S-T | 1024 | 8 | 50Hz | 4k | 2.66 | 0.92 | 0.59 | 7.07 | 0.84 |
| Encodec-2 | 1024 | 2 | 75Hz | 1.5k | 1.56 | 0.94 | 0.23 | 4.45 | **0.90** |
| DAC-2 | 1024 | 2 | 75Hz | 1.5k | 1.51 | 0.83 | 0.12 | 3.36 | 0.49 |
| BigCodec | 8192 | 1 | 80Hz | 1.04k | 2.68 | 0.93 | - | - | 0.84 |
| Xcodec | 1024 | 2 | 50Hz | 1k | 2.33 | 0.87 | - | - | 0.72 |
| S-T | 1024 | 2 | 50Hz | 1k | 1.25 | 0.77 | 0.68 | 8.02 | 0.36 |
| Mimi | 2048 | 8 | 12.5Hz | 1.1k | 2.24 | 0.90 | - | - | 0.73 |
| MBCodec | 2048 | 8 | 25Hz | 2.2k | 2.98 | 0.94 | 0.17 | 3.62 | 0.87 |
| S3Codec | 4096 | 8 | **12.5Hz** | 1.2k | 2.85 | **0.94** | **0.12** | 4.01 | 0.89 |

The combination of the semantic transformer and the acoustic transformer can guide the generation of target audio through the understanding of text and speech modalities, which conforms to the objective laws of human speech production. Finally, the overall optimization objective Eq.2 can be detailed as:

$$\mathbb{P}(\mathbf{A}|\mathcal{T}, \tilde{\mathbf{A}}) = \prod_{t=1}^{L_{|S|}} \left[ \mathbb{P}(\mathbf{S}_t|\mathbf{S}_{<t}, \mathcal{T}, \tilde{\mathbf{A}}; \theta_{\mathcal{S}}) \cdot \prod_{k=0}^{K-1} \mathbb{P}(\mathcal{A}_t^k|\mathcal{A}_t^{<k}, \mathbf{S}_t; \theta_{\mathcal{A}}) \right]. \tag{7}$$

Consequently, the overall goal of training optimization objective is fourmulated as follows:

$$\mathcal{L}_{total} = \sum_{t=1}^{L_{|S|}} \left[ ||\mathbf{S}_t - \theta_{\mathcal{S}}(\mathbf{S}_{<t}, \mathcal{T}, \tilde{\mathbf{A}})||_2 - \sum_{k=0}^{K-1} \log \mathbb{P}(\mathcal{A}_t^k|\mathcal{A}_t^{<k}, \mathbf{S}_t; \theta_{\mathcal{A}}) \right]. \tag{8}$$

The mathematical derivation can be found in the Appendix D.

### 3.3 Masked Audio Parallel Inference

Due to the uncertainty of each token prediction, especially in speech generation, errors accumulate, which reduces the expressiveness of the generated speech. To address this challenge, inspired by (Chen et al., 2025), we introduce Masked Audio Parallel Scaling in the semantic generation module. Specifically, for each prompt token sequence, we duplicate it $P$ times and apply a masking strategy to speech tokens separately with a certain probability, resulting in a total of $P$ token sequences. The model then produces $P$ output sequences, and these $P$ candidates are weighted and summed with a learnable weight to produce the final output sequence. Formally, in our speech generation task, the discrete text token embeddings and audio embeddings will be concatenated, resulting in the input embeddings, denoted as $\mathbf{x} \in \mathbb{R}^{L_{in} \times D}$. Specifically, we denote our trained semantic transformer $\theta_{\mathcal{S}} : \mathbb{R}^{L_{in} \times D} \to \mathbb{R}^{L_{in} \times D}$, where $\theta$ is the parameter, $L_{in}$ is the length of input text and audio embeddings and $D$ is the model dimension, the final output is formulated in the following form:

$$\theta_{\mathcal{S}}^*(\mathbf{x}) = w_1 \theta_{\mathcal{S}}(\mathbf{z}_1) + w_2 \theta_{\mathcal{S}}(\mathbf{z}_2) + \cdots + w_P \theta_{\mathcal{S}}(\mathbf{z}_P), \tag{9}$$

where $P$ denotes the number of parallel streams, $\mathbf{z}_1, \cdots, \mathbf{z}_p$ are $P$ distinct mask transformations of $\mathbf{x}$, and $w_1, \cdots, w_P$ are adaptive-trained aggregation weights. More details can be found in the Appendix B.

## 4 Experiments

### 4.1 Audio Quantization and Reconstruction Analysis

S3Codec is trained on the subset of our amassed speech data. Implementation details are listed in Appendix E.

Table 2: Objective evaluation in the SeedTTS test datasets.

| Model | test-zh | | test-en | | test-hard | |
|---|---|---|---|---|---|---|
| | WER(%) ↓ | SIM ↑ | WER(%) ↓ | SIM ↑ | WER(%) ↓ | SIM ↑ |
| **NAR-involved Models** | | | | | | |
| MaskGCT | 2.27 | 0.774 | 2.62 | 0.714 | 10.27 | 0.748 |
| E2 TTS (32 NFE) | 1.97 | 0.730 | 2.19 | 0.710 | - | - |
| F5-TTS (32 NFE) | 1.56 | 0.741 | **1.83** | 0.647 | 8.67 | 0.713 |
| Seed-TTS | **1.12** | **0.796** | 2.25 | **0.762** | 7.59 | **0.776** |
| FireRedTTS | 1.51 | 0.635 | 3.82 | 0.460 | 17.45 | 0.621 |
| CosyVoice | 3.63 | 0.723 | 4.29 | 0.609 | 11.75 | 0.709 |
| CosyVoice 2 | 1.45 | 0.748 | 2.57 | 0.652 | 6.83 | 0.724 |
| CosyVoice 3-0.5B | 1.16 | 0.780 | 2.02 | 0.718 | 6.08 | 0.758 |
| **Pure AR based Models** | | | | | | |
| QTTS | 1.66 | 0.648 | 3.17 | 0.652 | 14.45 | 0.641 |
| Spark-TTS | **1.20** | 0.672 | **1.98** | 0.584 | - | - |
| Llasa-1B-250k | 1.89 | 0.668 | 3.22 | 0.572 | 12.13 | 0.638 |
| Llasa-3B-250k | 1.60 | 0.675 | 3.14 | 0.579 | 13.37 | 0.652 |
| Llasa-8B-250k | 1.59 | **0.684** | 2.97 | 0.574 | 11.09 | 0.660 |
| CaT-TTS | 1.56 | 0.678 | 2.35 | **0.668** | **9.75** | **0.674** |

**Baselines.** To assess the reconstruction performance of S3Codec, we employ several state-of-the-art neural codecs as baselines, including Encodec (Défossez et al., 2022), DAC (Kumar et al., 2023), QDAC (Han et al., 2025), SpeechTokenizer (Zhang et al., 2023a), BigCodec (Xin et al., 2024), Xcodec (Ye et al., 2025a) and MBCodec (Zhang et al., 2025).

**Evaluation Metrics.** To evaluate the performance of S3Codec, we employ several metrics, including SIM, STFT Distance, Mel Distance, short-time objective intelligibility (STOI) (Taal et al., 2010) and perceptual evaluation of speech quality (PESQ) (Rix et al., 2001). All evaluations were conducted on the LibriSpeech (Panayotov et al., 2015) test-clean subset. More detailed evaluation set up is listed in Appendix E.2.

**Evaluation Results.** As shown in Table 4, S3Codec achieves SOTA-comparable performance with a very low frame rate in most evaluation dimensions. S3codec achieves higher SIM scores than MBcodec, Mimi, and SpeechTokenizer with the same codebooks. In terms of the restoration and perception indictors PESQ and STOI, S3codec is comparable to the high bitrates Encodec and DAC-8. At the evaluation dimension of STFT and Mel indicators, S3Codec also performs well among low-bitrate codecs. These results provide preliminary evidence of the model's effectiveness in reconstructing speech. As for the semantic evaluation, results in the Appendix E.2 demonstrate the superiority of S3Codec.

## 4.2 ZERO-SHOT TTS PERFORMANCE

**Datasets.** To train the CaT-TTS models, we have amassed a considerable dataset comprising multiple languages. The dataset contains about 200k hours labeled speech, with about 85% Chinese data and 15% English data. We evaluate our zero-shot TTS models with five benchmarks: (1) Seed-TTS test-en, a test set introduced in Seed-TTS of sample extracted from English public corpora, includes 1,000 samples from the Common Voice dataset. (2) SeedTTS test-zh, a test set introduced in Seed-TTS of samples extracted from Chinese public corpora, includes 2,020 samples from the DiDiSpeech (Guo et al., 2021) dataset. (3) Seed-TTS test-hard, includes 400 samples that consist of complex Chinese sentences. (4) PGC-Hard, includes 1500 Chinese samples, containing Professionally-Generated Content. (5) PGC-Poly, includes 1500 Chinese samples, containing polyphonic characters. The PGC testset is specially designed to test model generalization on difficult, out-of-domain voices.

**Evaluation Metrics.** We adopt the word error rate (WER) and speaker similarity (SIM) metrics for objective evaluation. For WER, we employ Whisper-large-v3 (Radford et al., 2023) and Paraformer-zh (Gao et al., 2023) as the automatic speech recognition engines for English and Mandarin, respectively. For SIM, we use WavLM-large fine-tuned on the speaker verification task to obtain

Table 3: Objective evaluation on hard mandarin test. $^\dagger$ represents the self-implemented model. $-$ means the average evaluation results across three sets.

| Model | Model Size | WER(%) ↓ | | | SIM↑ | UTMOS↑ |
| | | Seed-Hard | PGC-Hard | PGC-Poly | - | - |
|---|---|---|---|---|---|---|
| CosyVoice | 0.3B | 11.75 | 7.86 | 16.22 | 0.709 | 3.01 |
| CosyVoice 2 | 0.5B | **6.83** | **6.11** | 14.25 | **0.713** | 3.02 |
| L-CosyVoice50$^\dagger$ | 0.2B | 9.52 | 8.15 | 18.71 | 0.691 | 2.92 |
| L-CosyVoice25$^\dagger$ | 0.5B | 7.46 | 6.83 | **13.84** | 0.706 | 2.99 |
| Q-TTS | 0.2B | 14.45 | 7.89 | 14.37 | 0.654 | 3.03 |
| VALL-E$^\dagger$ | 0.2B | 13.12 | 9.68 | 15.71 | 0.631 | 3.05 |
| CaT-TTS | 0.4B | 9.75 | 7.03 | 13.97 | 0.672 | **3.13** |

speaker embeddings used to calculate the cosine similarity of speech samples of each test utterance against reference clips. For naturalness, we use SpeechMOS MOS prediction model to calculate UTMOS (Saeki et al., 2022) scores for evaluation.

**Baselines.** We compare our models with state-of-the-art zero-shot TTS systems, including Seed-TTS (Anastassiou et al., 2024), FireRedTTS (Guo et al., 2024), MaskGCT (Wang et al., 2024), E2 TTS (Eskimez et al., 2024), F5-TTS (Chen et al., 2024b), CosyVoice (Du et al., 2024a), CosyVoice2 (Du et al., 2024b), VALL-E (Wang et al., 2023) and QTTS (Han et al., 2025). Details of each model can be found in the Appendix F.2. In particular, we also compare the performance of SOTA two-stage models, including VALL-E, CosyVoice, CosyVoice 2, QTTS and self-implement AR (Llama) (Dubey et al., 2024) + flow-matching models (Lipman et al., 2022), where L-CosyVoice50 means Llama backbone with 50 Hz semantic codec (Hsu et al., 2021) and L-CosyVoice25 means with 25 Hz.

**Training.** We train CaT-TTS on 8 NVIDIA H20 96GB GPUs. The parallel stream is set to 4. For more details about the model architecture, please refer to Appendix F.1. We optimize the model with the AdamW optimizer with a learning rate of 1e-5 and 20K warm-up steps.

**Evaluation Results.** To evaluate CaT-TTS's zero-shot TTS capatility, we assess its performance on Seed-TTS-eval and compare it with existing zero-shot TTS models. These experiments focus on cross-sentence speaker similarity and the generation of intelligible speech. The results are presented in Table 4.1. As can be seen, CaT-TTS demonstrates a significant superiority in intelligibility for zero-shot TTS scenarios. With WER $1.56\%$, $2.35\%$ and $9.75\%$ in test-zh, test-en and test-hard, respectively, CaT-TTS achieves best or best comparable performance among these baselines, especially in pure AR based models. In terms of speaking similarity, like the other Pure AR based models, Cat-TTS's performance is inferior to NAR-involved models, especially pure NAR models. The reason is that NAR models like F5-TTS generate based on more explicit acoustic features like Mel-Spectrogram, and AR+NAR models typically construct acoustic information with acoustic guidance like speaker similarity vector in the NAR stage. Although with higher indicator performance, we think it may degrade diversity and cause more storage and processing cost during training. To do a further comprehensive comparison on zero-shot TTS performance, we compared recent prominent AR-based two-stage TTS models including VALL-E, CosyVoice, QTTS and reproduced Llama-CosyVoice as baseline models, and testify the synthesis capability in a more real scenarios. The evaluation datasets including PGC-Hard and PGC-Poly, which contain more complex real-life sentences and polyphonetic characters, respectively. The results in Table 4.2 demonstrate that CaT-TTS has SOTA comparable in-context learning ability. With WER $9.75\%$, $7.03\%$ and $13.97\%$ in Seed-TTS test-zh-hard, PGC-Hard and PGC-Poly, respectively. Q-TTS and VALL-E are Transformer-based TTS systems powered by codec, which is similar to CaT-TTS. As can be seen, CaT-TTS achieves better performance. Although without additional acoustic information supplement through flow-matching, CaT-TTS has comparable or superior performance in terms of UTMOS and WER, demonstrating the context-learning ability of our system.

### 4.3 ABLATION STUDY

**Modality UnderStanding.** To demonstrate the effectiveness and superiority of the modality understanding loss. We trained two models in sub-dataset with the same architecture but with small model

Table 4: Objective Evaluation. Comparison between models trained with and without semantic guidance.

| Model | WER(%) ↓ | | | SIM↑ | UTMOS ↑ |
|---|---|---|---|---|---|
| | SeedTTS-test | PGC-Hard | PGC-Poly | - | - |
| CaT-TTS w/o | 3.97 | 11.83 | 18.34 | 0.649 | 2.64 |
| CaT-TTS | 3.31 | 9.74 | 16.57 | 0.658 | 2.78 |

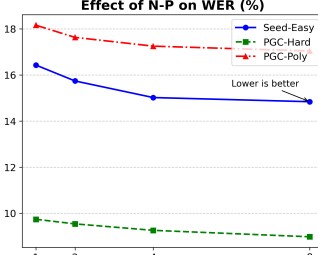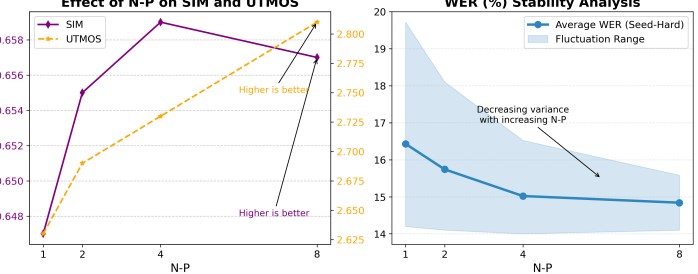

Figure 3: The result analysis of number of parallel streams.

size, and one of them is trained without semantic guidance. Table 4.2 shows the comparison results. With the loss of semantic guidance removed, this leads to performance decreases, especially with the WER increasing from $3.31\%$ to $3.97\%$ in SeedTTS-test, $9.74\%$ to $11.83\%$ in PGC-Hard and $16.57\%$ to $18.34\%$ in PGC-Poly, and the speech quality indicators SIM and UTMOS have also been reduced. During model training, semantic loss forces the semantic transformer to enhance its understanding of text and semantic modalities, thus improving the linguistic understanding ability of CaT-TTS. These results underscore the pivotal role of semantic loss in ensuring accurate semantic information learning, which is essential for maintaining high-fidelity generation of acoustic transformer.

**Masked Audio Parallel Inference.** To evaluate the effectiveness of masked parallel inference, we trained CaT-TTS-small in the subset of the collected dataset. We set different parallel streams and evaluated the performance in the PGC-Hard, PGC-Poly, and SeedTTS test-zh-easy dataset. Results in Figure 4 show the average performance analysis of MAPI parallel streams. The left subfigure shows the speech intelligibility improvement that MAPI brings. The middle subfigure shows that as the number of parallel streams increases, the acoustic performance SIM score and the UTMOS score show an upward trend. To demonstrate the robustness of MAPI, each sample in these datasets will be evaluated 10 times. As can be seen in the right subfigure in Figure 4, the performance of each inference is more stable in terms of the WER indicator. Due to the parallel computing capability of GPU, MAPI almost does not bring additional time consumption, but as the number of parallel streams increases, the utilization of GPU resources also increases. It is necessary to select the most appropriate number of parallel streams according to the requirements of the actual scenario.

## 5 CONCLUSION

In this work, we introduced CaT-TTS, a novel Text-to-Speech system designed to address key challenges in representation and generation. At its core is S3Codec, a split RVQ codec that resolves the trade-off between reconstruction fidelity and semantic interpretability by injecting linguistic features via ASR-based distillation. Building on this semantically aware representation, we proposed a principled "Understand-then-Generate" paradigm, realized through a dual-Transformer architecture that decouples contextual comprehension from acoustic rendering. To complement this, we developed Masked Audio Parallel Inference (MAPI), a nearly parameter-free inference strategy that enhances generation stability by dynamically mitigating local decoding errors. Extensive experiments demonstrate that the synergy between our architecture and codec allows CaT-TTS to achieve state-of-the-art performance in zero-shot voice cloning, with MAPI providing a measurable boost in robustness on benchmark datasets.

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

## A   THE USE OF LARGE LANGUAGE MODELS

We acknowledge the use of large language models (LLMs), such as OpenAI's ChatGPT, as a writing-assistance tool during the preparation of this manuscript. The primary use of the LLM was for improving the clarity and readability of the text, correcting grammatical errors, and rephrasing sentences. We emphasize that the LLM was used solely for text editing and was not involved in the generation of core scientific ideas, experimental design, data analysis, or the drawing of conclusions. All intellectual content, arguments, and the final manuscript were produced by the human authors, who take full responsibility for them.

## B   IMPLEMENTATION DETAILS OF MAPI

**Input Transformation** We expect that the transformations applied to the input embedding $x$ can significantly influence the output, which avoids excessively similar outputs across different parallel streams. Inspired by (Chen et al., 2025), we utilize random mask strategy to implement input transformation. To be specific, we first duplicate the input $\mathbf{x}$ into $P$ parallel copies, distinguishing them with different mask segments in each attention layer, which is sufficient to ensure diverse outputs across different streams.

**Output Aggregation** As stated in (Chen et al., 2025), dynamic aggregation weights performs better than static ones. Similarly, we concatenate each output together and use an MLP $h : \mathbb{R}^{d \times P} \to \mathbb{R}^P$ to convert it into a vector of length $P$ as aggregation weights. The process can be formalized as:

$$w_1, \cdots, w_P \leftarrow \text{Softmax}(h(\text{Concat}[\theta_\mathcal{S}(\mathbf{z}_1); \cdots ; \theta_\mathcal{S}(\mathbf{z}_P)])), \qquad (10)$$

where Softmax ensures aggregation weights are normalized, $\mathbf{z}_i$ represents masked input tokens. It can be seen as dynamically weighting different parallel streams during forward process for each token.

## C   MODEL ARCHITECTURE AND TRAINING RECIPE

### C.1   MODEL ARCHITECTURE AND SETTING

**Model Architecture** Our proposed audio codec is a fully convolutional autoencoder consisting of an encoder, a Residual Vector Quantizer (RVQ), and a decoder. The fundamental component of our architecture is a residual block, which contains a strided convolution for dimensionality change (downsampling or upsampling) followed by a stack of convolutional layers. We utilize the non-linear Snake function as the activation throughout these blocks. The encoder is composed of five such blocks, which progressively downsample the input waveform with strides of [2, 4, 5, 6, 8]. The decoder mirrors this structure with five corresponding upsampling blocks with strides of [8, 6, 5, 4, 2] and is configured with an internal channel dimension of 2048.

**Model Setting** To train the model, we employ a GAN-based objective with a combination of two discriminators: a multi-period discriminator [18] with periods of [2, 3, 5, 7, 11], and a complex multi-scale STFT discriminator. The STFT discriminator operates on three resolutions with window lengths [2048, 1024, 512] and a hop length of 1/4 the window size, using frequency band splits of [0.0, 0.1, 0.25, 0.5, 0.75, 1.0]. The total loss function is a weighted sum of a GAN loss, feature matching loss, a codebook loss, and a multi-resolution reconstruction loss. The reconstruction loss is computed as the L1 distance between the log-mel spectrograms of the original and reconstructed audio over seven different resolutions. These resolutions use window lengths of [32, 64, 128, 256, 512, 1024, 2048] with a corresponding number of mel bands [5, 10, 20, 40, 80, 160, 320], respectively.

### C.2   TRAINING OBJECTIVE

Our model is trained with a multi-task objective that jointly optimizes for reconstruction fidelity and semantic alignment. The primary task is reconstruction, which is guided by a GAN-based objective comprising a reconstruction term, a discriminative loss, and an RVQ commitment loss. This is complemented by a semantic distillation task, which introduces an additional loss term to ensure

the model's representations are linguistically meaningful. In the following, $\mathbf{x}$ represents an speech signal and $\hat{\mathbf{x}}$ denotes the reconstructed signal.

**Reconstruction Loss** The reconstruction loss comprises a time and a frequency domain loss. For time domain, we minimize the $L1$ distance between $\mathbf{x}$ and $\hat{\mathbf{x}}$, i.e. $\mathcal{L}_t = ||\mathbf{x} - \hat{\mathbf{x}}||_1$. For frequency domain, we use the $L1$ loss over the mel-spectrogram using several time scales. Formally, $\mathcal{L}_f = \sum_{i \in e} ||\mathcal{S}_i(\mathbf{x}) - \mathcal{S}_i(\hat{\mathbf{x}})||_1$, where $\mathcal{S}_i$ is a 64-bins mel-spectrogram using a normalized STFT with window size of $2^i$ and hop length of $2^i/4, e = 5, \cdots, 11$ is the set of scales.

**Discriminator Loss** We use the same discriminator as (Kumar et al., 2023) that consist of three discriminators. The adversarial loss is used to promote perceptual quality and it is defined as a hinge loss (Lim & Ye, 2017) over the logits of the discriminator, averaged over multiple discriminators and over time. Let $K$ denote the number of discriminators. For discriminators, $\mathcal{L}_D$ is defined as :

$$\mathcal{L}_D = \frac{1}{K} \sum_{k=1}^{K} \max(1 + D_k(\hat{\mathbf{x}}), 0) + \max(1 - D_k(\mathbf{x}), 0). \tag{11}$$

The adversarial loss for the generator $\mathcal{L}_g$ is constructed as follows:

$$\mathcal{L}_g = \frac{1}{K} \sum_{k=1}^{K} \max(1 - D_k(\hat{\mathbf{x}}), 0). \tag{12}$$

Additionally, the feature matching loss for the generator is computed as follow:

$$\mathcal{L}_{feat} = \frac{1}{KL} \sum_{K=1}^{K} \sum_{l=1}^{L} \frac{||D_k^l(\mathbf{x}) - D_k^l(\hat{\mathbf{x}})||_1}{mean(||D_k^l(\mathbf{x})||_1)}, \tag{13}$$

where the mean is computed over all dimensions and $L$ is the number of layers in discriminators.

**RVQ Commitment Loss** We add a commitment loss $\mathcal{L}_w$ between the pre-quantized value, and its quantized value, without gradient computed for the quantized value. The commitment loss is defined as : $\mathcal{L}_w = \sum_{i=1}^{N_q} ||\mathbf{z}_i - \mathbf{z}_{q_i}||$, where $\mathbf{z}_i$ and $\mathbf{z}_{q_i}$ denote current residual and nearest entry in the corresponding codebook respectively.

The generator is trained to optimize the following loss:

$$\mathcal{L}_G = \lambda_t \mathcal{L}_t + \lambda_f \mathcal{L}_f + \lambda_g \mathcal{L}_g + \lambda_{feat} \mathcal{L}_{feat} + \lambda_w \mathcal{L}_w + \lambda_{distill} \mathcal{L}_{distill}, \tag{14}$$

where $\lambda_{all}$ are the hyper-parameters used to balance each loss item. The detailed values are refered to (Kumar et al., 2023). $\lambda_{distill}$ is set to 0.1 in our work, and $\mathcal{L}_{distill}$ has been described in Section 3.1.1.

# D  TRAINING OBJECTIVE OF CAT-TTS

We use the maximum likelihood function to solve this problem.

$$\mathcal{L}_{total} = -\log \mathbb{P}(\mathbf{A}|\mathcal{T}, \tilde{\mathbf{A}})$$

$$= -\log \prod_{t=1}^{L_{|S|}} \left[ \mathbb{P}(\mathbf{S}_t|\mathbf{S}_{<t}, \mathcal{T}, \tilde{\mathbf{A}}; \theta_{\mathcal{S}}) \cdot \prod_{k=0}^{K-1} \mathbb{P}(\mathcal{A}_t^k|\mathcal{A}_t^{<k}, \mathbf{S}_t; \theta_{\mathcal{A}}) \right]$$

$$= -\sum_{t=1}^{L_{|S|}} \log \left[ \mathbb{P}(\mathbf{S}_t|\mathbf{S}_{<t}, \mathcal{T}, \tilde{\mathbf{A}}; \theta_{\mathcal{S}}) \cdot \prod_{k=0}^{K-1} \mathbb{P}(\mathcal{A}_t^k|\mathcal{A}_t^{<k}, \mathbf{S}_t; \theta_{\mathcal{A}}) \right]$$

$$= -\sum_{t=1}^{L_{|S|}} \left[ \log \mathbb{P}(\mathbf{S}_t|\mathbf{S}_{<t}, \mathcal{T}, \tilde{\mathbf{A}}; \theta_{\mathcal{S}}) + \log \prod_{k=0}^{K-1} \mathbb{P}(\mathcal{A}_t^k|\mathcal{A}_t^{<k}, \mathbf{S}_t; \theta_{\mathcal{A}}) \right]$$

$$= -\sum_{t=1}^{L_{|S|}} \left[ \log \mathbb{P}(\mathbf{S}_t|\mathbf{S}_{<t}, \mathcal{T}, \tilde{\mathbf{A}}; \theta_{\mathcal{S}}) + \sum_{k=0}^{K-1} \log \mathbb{P}(\mathcal{A}_t^k|\mathcal{A}_t^{<k}, \mathbf{S}_t; \theta_{\mathcal{A}})) \right]$$

$$= \sum_{t=1}^{L_{|S|}} \left[ -\log \mathbb{P}(\mathbf{S}_t|\mathbf{S}_{<t}, \mathcal{T}, \tilde{\mathbf{A}}; \theta_{\mathcal{S}}) - \sum_{k=0}^{K-1} \log \mathbb{P}(\mathcal{A}_t^k|\mathcal{A}_t^{<k}, \mathbf{S}_t; \theta_{\mathcal{A}})) \right]. \tag{15}$$

To be noticed, in Equation 5, we have

$$\mathcal{L}_{ctx} = -\sum_{t=1}^{L_{|S|}} \log \mathbb{P}(\mathbf{S}_t|\mathcal{T}, \mathbf{S}_{<t}, \tilde{\mathbf{A}}; \theta_{\mathcal{S}}) \rightarrow \mathcal{L}_{ctx} = \sum_{t=1}^{L_{|S|}} ||\mathbf{S}_t - \theta_{\mathcal{S}}(\mathbf{S}_{<t}, \mathcal{T}, \tilde{\mathbf{A}})||_2, \quad (16)$$

thus the above equation can be transformed as follows:

$$\mathcal{L}_{total} = \sum_{t=1}^{L_{|S|}} \left[ -\log \mathbb{P}(\mathbf{S}_t|\mathbf{S}_{<t}, \mathcal{T}, \tilde{\mathbf{A}}; \theta_{\mathcal{S}}) - \sum_{k=0}^{K-1} \log \mathbb{P}(\mathcal{A}_t^k|\mathcal{A}_t^{<k}, \mathbf{S}_t; \theta_{\mathcal{A}})) \right]$$

$$= \mathcal{L}_{total} = \sum_{t=1}^{L_{|S|}} \left[ ||\mathbf{S}_t - \theta_{\mathcal{S}}(\mathbf{S}_{<t}, \mathcal{T}, \tilde{\mathbf{A}})||_2 - \sum_{k=0}^{K-1} \log \mathbb{P}(\mathcal{A}_t^k|\mathcal{A}_t^{<k}, \mathbf{S}_t; \theta_{\mathcal{A}}) \right]. \quad (17)$$

# E    SEMANTIC SUPERIORITY OF S3CODEC

## E.1    DETAILS OF S3CODEC

To discretize waveforms into audio tokens, we introduce S3Codec, a neural audio codec that operates as an autoencoder with a discrete bottleneck. As Figure 2 shows, S3Codec consists of an autoencoder and Residual Vector Quantizer. Based on the DAC architecture (Kumar et al., 2023), the encoder projects a single-channel waveform $\mathbf{x} \in \mathbb{R}^T$ to a latent representation $\mathbf{A} = \text{enc}(\mathbf{x}) \in \mathbb{R}^{L \times D}$ by cascading residual convolutional blocks that interleave dilated and strided convolutions along with Snake nonlinearities and weight normalizaton, and Quantizer quantize the latent representation to disrete representations $\mathbf{C} \in \mathbb{R}^{K \times L \times D}$ where $L$ represents the length of encoded tokens, $K$ represents the number of codebooks and $D$ represents the dimension of codebook. Similarly to SpeechTokenizer and Mimi, we distill semantic information into the first level of RVQ. However, instead of using SSL models like HuBERT (Hsu et al., 2021) as a semantic teacher, we adopt Whisper (Radford et al., 2023), a state-of-the-art model for automatic speech recognition and speech translation whose hidden representation contains rich explicit linguistic features. It projects a 16kHz waveform into 1280-dimensional embeddings sampled at 50Hz, while S3Codec projects a 24kHz waveform into 4096-dimensional at 12.5 Hz. During training, we thus downsample the waveforms and project them to the same dimension as targets for distillation. Mimi (Défossez et al., 2024) found that, while distillation significantly improves the phonetic discriminability of the first quantizer, it also negatively affects the audio quality. To address the issue, we split the RVQ layers in a way similar to Mimi. Rather than a single RVQ with $K$ levels, we distill the semantic information into a plain VQ and apply an RVQ with $K - 1$ levels in parallel. Their outputs will be summed up; thus the constraint of acoustic information being conserved in the residual of the semantic quantizer is removed.

**Training Loss.**    We compute the frequency domain reconstruction loss using L1 loss on multi-scale mel-spectrograms. Multi-period discriminator and multi-band multi-scale STFT discriminator are used for waveform discrimination and frequency domain discrimination, respectively. RVQ codebook learning incorporates both a codebook loss and a commitment loss.

**Training Configuration.**    All audio samples are 24kHz. The codec has 8 codebooks, each with 4096 entries. For optimization, we use AdamW optimizer with moving average coefficients $\beta_1 = 0.8$ and $\beta_2 = 0.99$. The model converges within approximately 900k training steps using a batch size of 128.

**Evaluation Setup.** To evaluate the preservation of acoustic information, we employ several metrics. Speaker similarity (SIM) is calculated as the cosine similarity between speaker embeddings extracted from original and reconstructed audio using a pre-trained speaker verification model. STFT and Mel represent the spectrogram distance between original and reconstructed speech. We also use short-time objective intelligibility (STOI) (Taal et al., 2010) to measure speech intelligibility and perceptual evaluation of speech quality (PESQ) (Rix et al., 2001) to assess audio quality. All evaluations were conducted on the LibriSpeech (Panayotov et al., 2015) test-clean subset. To demonstrate the semantic alignment, we trained small CaT-TTS models powered by S3Codec and DAC, respectively.

Mel-Spectrogram Reconstruction Comparison

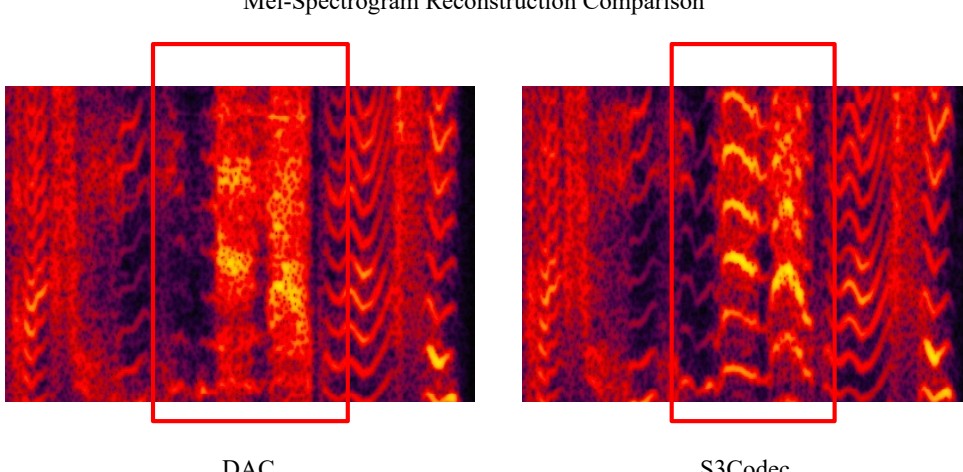

DAC                                    S3Codec

Figure 4: The result analysis of mel-spectrogram reconstruction.

## E.2 SEMANTIC PRESERVATION OF S3CODEC

To demonstrate the capability of semantic preservation of S3Codec, we trained CaT-TTS small powered with S3Codec and DAC, respectively. We use WER as the evaluation metric, representing the speech intelligibility of the generated results. Table 5 shows the evaluation results on Seed-TTS test zh easy, PGC-Hard and PGC-Poly. Compared to S3Codec-based system, DAC-based model's performance on speech intelligibility has decreased. The reason lies that DAC dose not contain structured linguistic features as S3Codec, which makes the LM model harder to understand, leading to worse performance than S3Codec.

| Model | SeedTTS-test | PGC-Hard | PGC-Poly |
|---|---|---|---|
| DAC-Based | 4.21 | 12.83 | 19.27 |
| S3Codec-Based | 3.30 | 9.75 | 16.53 |

Table 5: Objective Word Error Rate evaluation.

We visualize the mel-spectrogram reconstruction results below. As can be seen, the reconstruction results of S3Codec is more clear, while there exists a blurry segment in the result reconstructed by DAC.

## F IMPLEMENTATIONS OF CAT-TTS AND BASELINE DETAILS

### F.1 CAT-TTS ARCHITECTURE

**Semantic Transformer** Semantic Transformer is a decoder-only transformer. The dimension is 1536, with 12 layers.

**Acoustic Transformer** Acousctic Transformer is also a decoder-only architecture, with 8 layers, and the dimension is 1024.

**Text Tokenizer** We use the Whisper Tokenizer, with 50260 text vocabularies size.

Regarding the CaT-TTS small, the semantic transformer is 8 decoder-only transformer layers, with 1024 model dimension, and the acoustic transformer is 4 decoder-only transformer layers, with 512 model dimension.

## F.2 BASELINE DETAILS

**VALL-E (Wang et al., 2023):** AR + NAR TTS system. The first AR model predicts the first codebook, and the second transformer predict the remaining codebooks.

**Seed-TTS (Anastassiou et al., 2024):** Hybrid TTS system. A two-stage model that employs an AR LM for semantic token prediction and flow matching for acoustic feature generation.

**FireRedTTS (Guo et al., 2024):** Hybrid TTS system. A two-stage model similar to Seed-TTS, using an AR LM for semantic tokens and flow matching for acoustic features.

**MaskGCT (Wang et al., 2024):** NAR TTS system. A NAR model that applies masking-based generative strategies for speech synthesis.

**E2-TTS (Eskimez et al., 2024):** NAR TTS system. A flow matching-based model that predicts Mel spectrograms as acoustic features.

**F5-TTS (Chen et al., 2024b):** NAR TTS system. A flow matching-based model that predicts Mel spectrograms as acoustic features.

**CosyVoice series (Du et al., 2024a;b; 2025):** Hybrid TTS system. AR for semantic prediction and flow-matching for acoustic feature generation.

**Spark-TTS (Wang et al., 2025b):** Single codebook Neural Audio Codec based Pure language TTS model. Powered by BiCodec and Qwen LLM.

**QTTS (Han et al., 2025):** Pure Codec based language audio model. A two-stage AR+AR model. RVQ-based two stage speech synthesis modeling.

**Llasa (Ye et al., 2025b):** A single-stream codecbased TTS model that uses a single AR language model for direct single-stream code prediction.

