# OpenReview forum: "Comprehend and Talk: Text to Speech Synthesis  via Dual Language Modeling"
_ICLR.cc/2026/Conference — ICLR 2026 Conference Withdrawn Submission_

### Official Review · Reviewer_mDeA · 2025-10-19

**Soundness:** 2
**Presentation:** 2
**Contribution:** 2
**Rating:** 2
**Confidence:** 5

**Summary:**

The authors propose CaT-TTS, a semantically-grounded framework for zero-shot text-to-speech synthesis (TTS). The core idea of CaT-TTS is to explicitly integrate linguistic structure into an RVQ-based audio codec. Specifically, CaT-TTS introduces S3Codec, a split RVQ codec that distills semantic information from an ASR model (Whisper) into a primary codebook, while parallel residual RVQ captures acoustic details. Two Transformer modules (Semantic and Acoustic Transformers) then autoregressively synthesize hierarchical acoustic tokens. Experimental results demonstrate that CaT-TTS achieves comparable performance to existing zero-shot TTS models, such as Spark-TTS.

**Strengths:**

There are two major strengths:

(1) Low frame rate is impressive. S3Codec operates at 12.5 Hz, which is competitive among recent neural audio codecs.

(2) Comparisons against SOTA TTS indicate effectiveness of , CaT-TTS particularly in UTMOS.

**Weaknesses:**

(1) Limited Novelty\
ASR-based distillation has been extensively studied in neural audio codecs (NACs). Although the proposed split RVQ introduces some novelty, results presented in Table 1 do not demonstrate clear advantages over recent NACs. Specifically, performance metrics such as PESQ and Mel distance show notable degradation when compared to DAC-8 and MBCodec.

(2) Fairness of Comparisons\
Key hyperparameters are inconsistently configured in Table 1. For example, the "CB" parameter (presumably indicating codebook size) is larger for S3Codec than for baseline NACs like Mini and MBCodec. Adjusting codebook size upward and frame rate downward for existing codecs could potentially narrow the observed performance gap. Therefore, broader and more balanced hyperparameter sweeps are required for both S3Codec and baseline models. Additionally, the absence of BiCodec from Table 1 (whereas Spark-TTS is included in Table 2) raises concerns about the comprehensiveness of the comparisons.

(3) Low Frame Rate Evaluation\
Evaluation at frame rates below 12.5 Hz is necessary. Recent literature, such as HALL-E (ICLR 2025), has reported evaluations at 8 Hz for the first RVQ layer. Evaluating CaT-TTS at or below 8 Hz would further strengthen the claims made in the paper.

(4) Ablation Studies\
Table 4 currently presents results solely with and without semantic guidance. Critical aspects, including the effectiveness of split RVQ and the specific selection of Whisper over alternative models such as HuBERT, are not independently analyzed.

**Questions:**

Why was evaluation limited to frame rates at 12.5 Hz? Could the authors also provide results at or below 8 Hz?

---

### Official Review · Reviewer_jTzu · 2025-10-30

**Soundness:** 3
**Presentation:** 3
**Contribution:** 1
**Rating:** 2
**Confidence:** 5

**Summary:**

The paper proposes CaT-TTS (Comprehend-and-Talk), an autoregressive text-to-speech framework that combines a dual-stage architecture with a semantically-aware audio codec. The model introduces S3Codec, a split residual vector quantization (RVQ) codec designed to capture both semantic and acoustic information by distilling features from a pretrained ASR model (Whisper) into the first quantizer. Built on top of S3Codec, the system uses two stacked transformers: a semantic transformer that models the relationship between text and high-level semantic speech tokens, and an acoustic transformer that generates fine-grained acoustic tokens conditioned on this representation.

To address error accumulation during autoregressive decoding, the authors propose a Masked Audio Parallel Inference (MAPI) strategy, which samples multiple masked variants of the semantic representation in parallel and aggregates their outputs to improve stability. Experiments show that the proposed system performs comparably to or slightly better than existing models in terms of speech quality, similarity, and intelligibility, while maintaining zero-shot voice cloning capability.

**Strengths:**

1. The paper is well-structured, with a logical progression from codec design to model architecture and inference.

2. The proposed dual-transformer framework (semantic + acoustic) is consistent with the natural separation between linguistic and acoustic modeling in speech synthesis.

3. The results are reasonable and within the expected range of current tokenized TTS systems.

4. The authors some analysis of the inference-time stability problem, attempting to mitigate it through the proposed MAPI strategy.

**Weaknesses:**

1. The overall novelty is limited. The main technical contributions—semantic distillation into the first codebook and the separation of semantic and acoustic transformers—are direct extensions of existing frameworks such as Mimi, SpeechTokenizer, and standard two-stage text-to-semantic-to-acoustic TTS pipelines.

2. S3Codec primarily replaces the SSL model used for distillation with an ASR encoder (Whisper), which does not constitute a significant conceptual advance.

3. The MAPI inference method appears to be a form of test-time augmentation by generating multiple masked inputs and averaging outputs. Similar effects could probably be achieved more efficiently by adjusting sampling temperature or using stochastic decoding.

4. The model being fully autoregressive, and the added inference strategy further increases computational cost, which might limit practical benefit.

5. The title and framing around “comprehension” are somewhat misleading; the system performs standard two-stage token generation rather than true semantic understanding.

6. The paper relies on private training data, and the provided anonymous code link is a placeholder, raising reproducibility and transparency concerns.

7. The results, while acceptable, do not clearly surpass strong existing baselines, reducing the paper’s empirical impact.

**Questions:**

See Weaknesses above.

---

### Official Review · Reviewer_Cx6b · 2025-11-01

**Soundness:** 3
**Presentation:** 2
**Contribution:** 1
**Rating:** 2
**Confidence:** 5

**Summary:**

This paper presents CaT-TTS, a zero-shot TTS that integrates a semantically-aware codec with a dual-Transformer architecture to separate understanding and generation. By leveraging semantic distillation and a masked audio parallel inference (MAPI) strategy for stable decoding.

**Strengths:**

The paper introduces Masked Audio Parallel Inference and shows that MAPI effectively enhances generation stability and speech quality with minimal additional latency.

**Weaknesses:**

- The major weakness is that the contribution of this paper appears incremental rather than novel. The overall architecture is largely similar to Moshi, and the proposed S3Codec, which modifies Mimi’s architecture with larger codebooks (from 2048 to 4096), shows only limited performance improvement. In addition, the performance of the proposed model is limited when compared to the baseline models.

- The so-called “Understand-then-Generate” paradigm in CaT-TTS is conceptually similar to previously established frameworks such as Thinker-Talker or Planner-Decoder architectures. Therefore, while the paper presents a well-structured integration, its novelty in architectural design is limited, as the dual-Transformer separation between semantic understanding and acoustic generation has already been explored in prior works.

- The paper points out the issue of degraded audio quality in Mimi’s split RVQ distillation and proposes a modified approach that performs semantic distillation separately on a plain VQ. However, Appendix E.2 only presents a comparison between S3Codec and DAC, showing that S3Codec preserves linguistic information more effectively. Since models employing semantic distillation are already known to retain semantic information better than standard RVQ-based codecs, this comparison is somewhat limited. Therefore, conducting a more thorough ablation study on the Mimi structure would strengthen the validity of the proposed modifications. Specifically, it would be important to analyze whether the observed improvement arises from (1) the difference between SSL and ASR teachers, (2) the benefit of applying distillation to a separate plain VQ, or (3) the trade-off between the increased codebook size and the resulting resource overhead.

**Questions:**

- It is unclear whether the Semantic Transformer was initialized from an existing LLM or trained from scratch. The model is reported to have 0.4B parameters—where does this number come from?

- The method increases GPU resource usage as the number of parallel streams grows — how does the model balance this trade-off in real-time or large-scale scenarios?

---

### Official Review · Reviewer_9G9x · 2025-11-02

**Soundness:** 3
**Presentation:** 3
**Contribution:** 2
**Rating:** 2
**Confidence:** 4

**Summary:**

This paper claims it provides three contributions.
+ Using ASR to make one layer of RVQ has moe content information (via distillation)
+ A two-stage decoding strategy: a semantic transformer and an acoustic transformer
+ Masked Audio Parallel Infer ence (MAPI).

**Strengths:**

The paper is easy to read. The proposed TTS model shows good performance compared with the baselines.

**Weaknesses:**

### 1. ASR-Guided Semantic Distillation into RVQ

The proposed ASR-guided semantic distillation into RVQ represents an incremental contribution rather than a major innovation. Previous works such as SpeechTokenizer guided the first RVQ layer with a semantic teacher (HuBERT), and Mimi also distilled semantic features into RVQ-1.
This paper instead employs ASR-derived supervision, which is a different, but not fundamentally new.

Furthermore, the idea of using ASR-based supervision to make one stream or token layer more “content-like” is not novel, either. For example:
- PAST ([arXiv:2505.14470](https://arxiv.org/abs/2505.14470)) supervised its first codebook using phonetic/ASR tasks.
- QTTS / QDAC ([arXiv:2507.12197](https://arxiv.org/abs/2507.12197)) explicitly employed an autoregressive ASR model to guide the first codebook.

Even if PAST and QTTS/QDAC are treated as concurrent works, the authors’ claim that *“ASR teacher is better than SSL teacher”* remains unsupported.  There is no ablation study that replaces Whisper with a strong SSL teacher for comparison, which weakens this claim.

### 2. Two-Stage Decoding Architecture

The two-stage decoding approach is also incremental. While the paper introduces the idea of predicting continuous “semantic embeddings” with an MSE loss as an intermediate representation, this component’s benefit is not empirically verified.  The paper lacks an ablation study demonstrating that this design improves performance compared with direct discrete prediction or other alternatives.

### 3. Evaluation and Analysis

There are no subjective listening tests, which are essential to validate perceptual quality in speech generation tasks.

**Questions:**

Please check the link you provided in your abstract. When I click the link to the demo page, I am unable to see the audio files.

Can we consider the two-stage decoding approach here as an encoder-decoder architecture? The difference is that the encoder here is causal (although it might also be possible to try a full attention encoder here).

---

### Note · Authors · 2025-11-12

I have read and agree with the venue's withdrawal policy on behalf of myself and my co-authors.